# Decoupling Vision and Reasoning: A Data-Efficient Pipeline for Surgical VQA

Mohamed Hamdy[1]                                                    MM1905748@QU.EDU.QA
Fatmaelzahraa Ali Ahmed[2]                                  FATMAAHMED.HMC@GMAIL.COM
Muraam Abdel-Ghani[2]                                   MURAAM.ABDELGHANI@OUTLOOK.COM
Muhammad Arsalan[1]                                       MUHAMMAD.ARSALAN@QU.EDU.QA
Ponnuthurai Nagaratnam Suganthan[1]                          P.N.SUGANTHAN@QU.EDU.QA
Khalid Al-Jalham[2]                                                KALJALHAM@HAMAD.QA
Abdulaziz Al-Ali[1]                                                   A.ALALI@QU.EDU.QA
Shidin Balakrishnan[2]                                       SBALAKRISHNAN1@HAMAD.QA

[1] Computer Science and Engineering Department, College of Engineering, Qatar University, Doha, Qatar

[2] Department of Surgery, Hamad Medical Corporation, Doha, Qatar

**Editors:** Accepted for publication at MIDL 2026

## Abstract

Vision-language models (VLMs) are becoming increasingly important for surgical intelligence, where reliable scene understanding requires combining visual perception with language-based reasoning. However, progress is constrained by the scarcity of high-quality multimodal datasets, making end-to-end training more prone to overfitting. Existing approaches often address this limitation by converting task-specific datasets (e.g., segmentation, phase recognition, tool-tissue interaction) into synthetic vision-question answering (VQA) form, but such conversions provide only sparse supervision and limit generalization. To overcome these challenges, we propose Surg-SAGE (Structured Abstraction from Granular Experts), a modular pipeline that decouples vision information extraction from reasoning. Specialist surgical models–proven effective for their corresponding vision tasks–are first used to extract task-relevant signals, which are then transformed via heuristics into structured textual descriptions. These descriptions, together with the clinical question, are passed to a large language model (LLM) that performs the reasoning step and provides the answer. The novelty of this work lies in demonstrating that decoupling perception from language processing and leveraging expert-trained specialist models enables strong VQA performance, even when paired with relatively lightweight, frozen LLMs and without requiring multimodal training data. We evaluate this pipeline on the EndoVis-18-VQA benchmark under different configurations of specialist models and LLMs, showing that combining complementary experts yields stronger performance than relying on any single model. Surg-SAGE achieves higher accuracy, recall and F1 than existing surgical VQA baselines, with improvements of up to 2.3% in accuracy without requiring multimodal training, establishing abstraction-driven modularity as a data-efficient and generalizable paradigm for surgical vision-language understanding.

**Keywords:** Surgical VQA, Modular Vision-Language Models, Vision Language Models, Multi-modal Reasoning

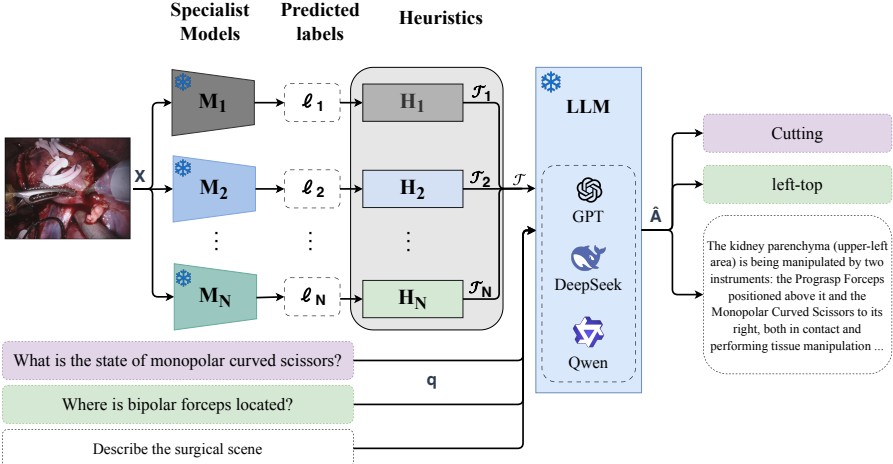

Figure 1: Overview of the proposed modular surgical VQA pipeline. Given an endoscopic frame $X$ and a question $q$, task-specific specialist models extract visual signals $\ell$ (e.g., segmentation and tool–tissue interactions), which are abstracted via deterministic heuristics into structured textual descriptions $\mathcal{T}$. These are provided to a frozen LLM to produce the answer $\hat{A}$. Snowflake icons denote frozen components.

## 1. Introduction

Minimally invasive surgery now underpins contemporary surgical practice due to their reduced incision size, postoperative pain, and recovery time (Sijberden et al., 2025). These procedures generate continuous, high-resolution endoscopic video that captures anatomy, instruments, and workflow, providing a rich substrate for computer vision and vision-language methods (Buia et al., 2015; Mascagni et al., 2022). Accordingly, much of the progress in surgical AI has been driven by task-specific vision models for tool and tissue segmentation (Allan et al., 2020; Hong et al., 2020), workflow phase recognition (Twinanda et al., 2016), tool-tissue interaction analysis (Islam et al., 2020; Seenivasan et al., 2022b), and higher-level action triplet recognition (Nwoye et al., 2022).

Despite their success, these models remain constrained by narrowly defined tasks such as segmentation, phase recognition, or interaction classification. To achieve broader reasoning capabilities, recent work has turned toward vision-language models (VLMs), which couple surgical perception with natural-language supervision. By aligning video with text, VLMs offer richer semantics, interpretable outputs, and more flexible clinical interaction. Large-scale efforts such as SurgVLM (Zeng et al., 2025) and SurgVLP (Yuan et al., 2025) illustrate the feasibility of training generalist surgical models, while systems like Surgical-GPT (Seenivasan et al., 2023) and Surgical-VQA (Seenivasan et al., 2022a; Yuan et al., 2024a) highlight the potential of answering clinically relevant queries directly from surgical scenes, including approaches based on scene-graph or neurosymbolic abstractions that learn or assume explicit symbolic representations and perform reasoning via graph inference or executable programs (Andreas et al., 2016; Johnson et al., 2017; Hudson and Manning, 2019; Hu et al., 2019; Mao et al., 2019).

A central challenge for surgical vision-language modeling is the dependence on large multimodal datasets, which remain scarce and costly to curate (Yuan et al., 2024b). Unlike natural-image domains with abundant paired corpora, surgical data requires expert annotation and is often limited to specific procedures. To expand supervision, prior works repurpose existing task-specific datasets—such as segmentation, workflow recognition, or tool-tissue interaction—into synthetic VQA form (Seenivasan et al., 2022a; Yuan et al., 2024a; Zeng et al., 2025). While effective, these conversions provide only narrow language supervision tied to single queries (Wang et al., 2024b; Menon and Vondrick, 2022) and can introduce noise when relying on automatically transcribed audio (Yuan et al., 2024c,b). These limitations motivate frameworks that reduce reliance on synthetic multimodal alignment and instead directly leverage existing expert-trained models.

In this work, we propose Surg-SAGE (Structured Abstraction from Granular Experts), a modular pipeline for surgical vision-language understanding that overcomes these limitations. Instead of training end-to-end multimodal models, we decouple perception from reasoning by relying on frozen, expert-trained vision models to extract task-specific information, leveraging the fact that such specialists are already highly effective for their respective vision tasks, which is then translated into structured textual descriptions for language-based inference. This separation offers two main advantages. First, it eliminates the need for large-scale multimodal supervision, avoiding both the inefficiency of synthetic VQA conversions and the noise of weakly aligned transcripts. By transforming visual predictions into human-readable text, the intermediate representations remain interpretable while reducing the risk of overfitting. Second, by incorporating multiple specialized models–such as segmentation and tool-tissue interaction networks–we aggregate complementary signals that together provide a richer and more clinically relevant description of the scene. By explicitly integrating different specialist models, their corresponding domain-specific skills are directly embedded into the pipeline, without requiring these capabilities to be learned through multimodal supervision.

Our contributions are three-fold:

- We introduce a general framework that abstracts surgical perception into textual descriptions, showing that effective surgical VQA can be achieved by decoupling perception from language processing without multimodal training.

- We define heuristics to convert outputs from specialist models into interpretable inputs for LLM, making the reasoning process transparent.

- We demonstrate through extensive experiments on EndoVis-18-VQA that combining complementary vision experts with LLM reasoning surpasses prior baselines, establishing abstraction-driven modularity as a data-efficient and generalizable paradigm for surgical VQA.

## 2. Problem Setup and Notation

We study the problem of surgical visual question answering (VQA), where the input is an endoscopic frame and a natural language question, and the goal is to return a clinically relevant answer. Our design departs from traditional end-to-end vision–language models

by decoupling perception from reasoning. Specifically, we factor the pipeline into three stages:(i) task-specific *specialist models* that extract visual information, (ii) *heuristics* that map these predictions into structured textual representations, and (iii) a frozen LLM that performs reasoning. This modular decomposition reduces the dependence on large-scale multimodal datasets, while maintaining interpretability and flexibility.

**Inputs.** Let $X \in \mathbb{R}^{H \times W \times 3}$ denote a surgical image, and let $q$ denote a natural language question. The objective is to predict an answer $\hat{A}$ expressed in natural language.

**Specialist models.** Previous research has shown that a single vision encoder typically captures only part of the information contained in visual scenes. For example, language-supervised encoders such as CLIP (Radford et al., 2021) align well with semantics but often miss fine-grained spatial cues, whereas self-supervised encoders like DINOv2 (Oquab et al., 2023) or segmentation experts like SAM (Kirillov et al., 2023) capture complementary information (Fan et al., 2024; Tong et al., 2024). Multi-expert systems that combine such models consistently improve representation quality and downstream performance (Shi et al., 2024). Motivated by these insights, we assume access to a collection of $N$ task-specific vision experts $\{M_i\}_{i=1}^{N}$, each trained to solve a distinct perception problem. Given an input image $X$, each expert produces a task-level prediction

$$\ell_i = M_i(X),$$

where the form of $\ell_i$ depends on the task—for instance, a categorical label, a dense segmentation mask, or a set of interaction triplets.

**Heuristics.** The outputs $\ell_i$ produced by the specialist models are heterogeneous, ranging from discrete labels to dense masks or structured tuples. Multimodal learning often fuses such outputs via trainable projection layers, requiring large-scale multimodal datasets, which are scarce in surgical domains. To avoid these limitations, we introduce a family of deterministic heuristics that abstract model predictions into symbolic textual descriptions, thus enabling reasoning entirely in the language domain while keeping the LLM frozen. Formally, for each task $i$ we define

$$H_i : \ell_i \mapsto \mathcal{T}_i,$$

where $\mathcal{T}_i$ is a set of textual statements describing $\ell_i$. Aggregating across all models yields

$$\mathcal{T} = \bigcup_{i=1}^{N} \mathcal{T}_i,$$

an intermediate representation for reasoning. These heuristics range from simple templates (e.g., interaction labels) to more complex rule-based procedures (e.g., spatial relations from segmentation masks).

**Language reasoning.** Once visual predictions are abstracted into text, the reasoning step is handled by an LLM. Unlike traditional multimodal systems requiring joint training to align image and text embeddings, we use a frozen LLM purely as an inference engine. This avoids additional multimodal supervision and leverages the broad reasoning capabilities of pretrained language models. Formally, given a frozen LLM, $q$ and $\mathcal{T}$ derived from heuristics, the model predicts an answer

$$\hat{A} = \text{LLM}(q, \mathcal{T}).$$

Operating exclusively on interpretable textual descriptions provides a flexible means of combining heterogeneous visual cues without requiring any gradient-based adaptation to the surgical domain as the perception burden is already addressed by the specialist models.

**Overall pipeline.** The complete formulation can thus be expressed as

$$X \xrightarrow{M_i} \ell_i \xrightarrow{H_i} \mathcal{T}_i \xrightarrow{\cup} \mathcal{T} \xrightarrow{\text{LLM}(q,\cdot)} \hat{A}.$$

An illustration of this general pipeline is provided in Figure 1.

## 3. Methodology

Our methodology instantiates the general pipeline described in Section 2 for the task of surgical VQA. We detail the concrete choices of specialist models, the design of heuristics for textual abstraction, and the language reasoning stage.

### 3.1. Specialist Model Choices

To cover essential aspects of surgical scene understanding, we rely on two classes of vision experts: *segmentation models* and *interaction models*, capturing complementary low-level spatial structure and high-level functional cues.

#### 3.1.1. SEGMENTATION

Segmentation provides dense spatial evidence–what structures are present, where they are located, and how elements are arranged. Without such structure, an LLM lacks the spatial grounding needed for clinically precise answers. Prior work suggests that segmentation masks alone can support the generation of clinically relevant descriptions of surgical scenes (Hamdy et al., 2025). Segmentation therefore supplies a dense, procedure-agnostic representation $S \in \{0, \ldots, C\}^{H \times W}$ that our heuristics convert into textual facts. We experiment with two models: FASL and a hybrid SAM+FASL variant.

**FASL.** The Feature-Adaptive Spatial Localization model (FASL) (Muraam et al., 2025) improves holistic surgical scene segmentation by combining low-level and high-level features. It achieves strong performance across instrument and anatomy segmentation benchmarks, surpassing prior baselines. These capabilities make FASL a suitable choice for providing the semantic scene information required by Surg-SAGE.

**SAM+FASL.** While specialist pixel-wise classifiers perform strongly on curated datasets, they may struggle under distribution shift or produce fragmented labels. Prior work suggests decoupling *mask generation* from *classification* via foundation segmenters followed by domain-specific classifiers (Hu et al., 2023; Wang et al., 2024c). Following this strategy, we construct a hybrid SAM+FASL model: SAM (Kirillov et al., 2023) produces region masks $\mathcal{R} = \{R_k\}$, while FASL generates a label map $S$. Each region is assigned the class

$$\hat{y}(R_k) = \arg \max_{c \in \{1, \ldots, C\}} \big| \{ p \in R_k \mid S(p) = c \} \big|,$$

that is, the class most frequently predicted by FASL within the region. This majority-vote labeling leverages SAM's generalization for region proposals while retaining FASL's domain-specific semantics for class assignment. A qualitative example of this process is shown in the Appendix Figure 4.

### 3.1.2. Interaction

Understanding tool-tissue interactions is a key step in surgical scene understanding, since it provides not only knowledge of *what* instruments and anatomical structures are present, but also *how* they functionally relate during a procedure, which is important for skill assessment, feedback, and decision support (Islam et al., 2020).

**Tool-tissue interaction.** We adopt the graph-based interaction model of (Seenivasan et al., 2022b), where instruments and tissues are nodes and functional relationships (e.g., grasping, cutting, retracting, suction, idle) are edges. Using both visual-semantic and relational reasoning, the model predicts the interaction state of each instrument-tissue pair.

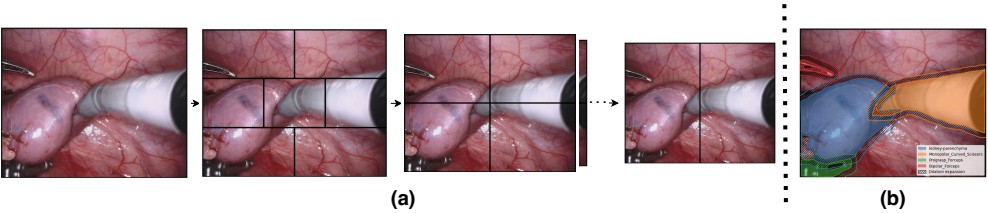

(a)  (b)

Figure 2: Illustration of segmentation-based heuristics used for textual abstraction. (a) Absolute location: object instances are assigned to symbolic spatial regions (e.g., top-left) using a prioritized hierarchy of partitions. (b) Dilation-based adjacency: spatial relationships and proximity between objects are inferred by progressively dilating segmentation masks to detect contact or closeness. These heuristics produce compact, interpretable textual facts for downstream language processing.

## 3.2. Heuristics Design

To instantiate the abstraction step, we apply deterministic heuristics that convert the heterogeneous predictions of the specialist models into structured textual descriptions. These rules bridge perception and reasoning by translating segmentation masks and interaction labels into interpretable, token-efficient statements that the LLM can consume.

### 3.2.1. Segmentation

Given a semantic segmentation mask $S$, the module $H_{\text{seg}}$ produces a fact set summarizing object presence, coarse layout, and spatial relationships. A high-level illustration of this process is shown in Figure 2. The module generates three families of statements:

**Absolute location.** Each object instance is assigned to a symbolic region (e.g., "top-left", "center") based on a small hierarchy of spatial partitions, enabling a coarse but stable description of where structures appear.

**Pairwise spatial relations.** When multiple objects occupy nearby regions, we infer simple directional relations (e.g., "the forceps is to the left of the needle") using adjacency cues and centroid orientation.

**Graded proximity.** For instrument-tissue pairs, we estimate discrete proximity levels (e.g., "touching", "near", "far") through successive neighborhood expansions around the instrument mask, providing soft cues about closeness.

Together, these heuristics yield $\mathcal{T}_{\text{seg}}$, a compact symbolic summary of the scene.

### 3.2.2. INTERACTION

The interaction model outputs categorical labels describing functional relations such as co-agulation, cutting, or idle. The heuristics module $H_{\text{int}}$ converts these predictions into concise declarative facts using simple templates, such as "monopolar curved scissors: cutting," forming the fact set $\mathcal{T}_{\text{int}}$. These statements complement the spatial information extracted from segmentation, enabling the LLM to reason about both *where* objects are and *how* they are being used.

Full algorithmic details for all heuristics are provided in Appendix A.

### 3.3. Language Reasoning

The fact sets $\mathcal{T}_{\text{seg}}$ and $\mathcal{T}_{\text{int}}$ are concatenated into a unified description $\mathcal{T}$. Given a question $q$, the frozen LLM predicts the answer $\hat{A}$ by reasoning over this structured textual summary.

## 4. Experimental Setup

**Datasets.** We evaluate Surg-SAGE on the test split of the **EndoVis-18-VQA** benchmark (Seenivasan et al., 2022a), which contains roughly 2.7k question–answer pairs. We opt for **EndoVis-18-VQA** since it captures two core capabilities fundamental to frame-wise surgical understanding: spatial understanding (via segmentation) and functional understanding (via instrument–tissue interaction). Specialist models are trained separately: the segmentation expert (FASL) uses the EndoVis-18 annotations (Allan et al., 2020), and the tool–tissue interaction expert is trained on the dataset of Islam et al. (2020). Because the EndoVis-18-VQA split differs from the partitions usually used for training perception specialists, we retrain FASL using only videos that do not appear in the VQA test set, holding out videos 1, 5, and 16 to prevent data leakage.

**Competing models.** We benchmark against representative surgical VQA baselines, including VisualBERT (Li et al., 2019), VisualBERT RM (Seenivasan et al., 2022a), and LV-GPT with RN18 and Swin backbones (Seenivasan et al., 2023). We also compare with Surgical-LVLM (Wang et al., 2024a) and recent surgical vision-language models such as SurgVLM (Zeng et al., 2025).

**Language models.** For the reasoning stage, we evaluate multiple frozen LLMs accessed via their official APIs. From OpenAI, we include GPT-4o, GPT-4o-mini, GPT-5, GPT-5-mini, and GPT-5-nano; from DeepSeek, we include DeepSeek-chat and DeepSeek-reasoner. We additionally test open-source vision-language models from the Qwen-VL family (2.5 and 3.0), including the 32B, 72B, and 8B instruct variants (Bai et al., 2025b,a), which are accessed via the OpenRouter API. All models operate on identical serialized textual fact sets to ensure fair comparison.

**Implementation details.** Specialist models are trained and evaluated on a workstation with an RTX 4090 GPU and Intel i9 CPU. FASL follows the training protocol of Muraam et al. (2025), while the interaction expert is loaded from a publicly available checkpoint.[1] LLMs are queried via external APIs, and are used without further fine-tuning. Evaluation metrics include overall accuracy, recall, and F1 score.

---

1. https://github.com/lalithjets/Global-reasoned-multi-task-model

Table 1: Performance of segmentation and interaction specialists on EndoVis-18-VQA. Rows are grouped into segmentation-only, interaction-only, and combined configurations. Columns under *Overall Performance* report Acc, Recall, and F1, while those under *Question Type (Acc)* report accuracy per question category (Organ, State, Location). 'GT' denotes oracle annotations (i.e., ground truth provided directly), and 'INT' refers to the tool-tissue interaction specialist model. Bold values indicate the best performance among non-oracle models. All results in the table are reported using GPT-4o-mini as the reasoning LLM.

| Seg. Model | Interaction | Overall Performance | | | Question Type (Acc) | | |
|---|---|---|---|---|---|---|---|
| | | Acc | Recall | F1 | Organ (16%) | State (42%) | Location (42%) |
| FASL | — | 0.6208 | **0.3674** | 0.3397 | **1.0000** | 0.4091 | 0.6865 |
| SAM+FASL | — | 0.6194 | 0.3354 | 0.3032 | **1.0000** | 0.4539 | 0.6382 |
| GT | — | 0.6677 | 0.5337 | 0.4570 | 1.0000 | 0.4109 | 0.7967 |
| — | INT | 0.4536 | 0.2039 | 0.1558 | **1.0000** | **0.6916** | 0.0052 |
| — | GT | 0.5778 | 0.5770 | 0.4274 | 1.0000 | 0.9905 | 0.0026 |
| GT | GT | 0.8996 | 0.8242 | 0.8304 | 1.0000 | 0.9543 | 0.8062 |
| GT | INT | 0.7862 | 0.4243 | 0.4336 | 1.0000 | 0.6693 | 0.8208 |
| FASL | GT | 0.8472 | 0.7497 | 0.7523 | 1.0000 | 0.9414 | 0.6942 |
| FASL | INT | **0.7299** | 0.3516 | **0.3605** | **1.0000** | 0.6615 | **0.6942** |

## 5. Results and Discussion

**Specialist models.** Table 1 summarizes performance across segmentation-only, interaction-only, and combined specialist configurations. Oracle rows (GT) provide upper bounds using ground-truth masks or interaction labels. Among single experts, segmentation clearly dominates, where FASL achieves 62% accuracy versus 45% for interaction-only predictions, indicating that spatial cues from segmentation masks enable LLMs to infer a broader range of clinically relevant answers than interaction labels alone. Combining segmentation and interaction consistently improves performance (FASL+INT: 73% accuracy, 36% F1), while GT+GT reaches 90% accuracy and 83% F1, quantifying the performance ceiling in the absence of perception errors.

Per-question analysis further highlights complementary strengths. Although the GT+GT oracle achieves perfect accuracy on organ and state questions, it still incurs occasional spatial errors due to ambiguous boundaries and the discretized answer space (e.g., coarse region choices such as "top-right" vs. "bottom-right"), rather than missing visual information. Segmentation variants excel on location-oriented questions (FASL: 69% accuracy) but struggle on state queries, whereas interaction models perform best on state questions (INT: 69%) while largely failing on location (0.5%). Combining both sources recovers the strengths of each, yielding more balanced performance across categories and confirming that segmentation and interaction specialists provide complementary signals for reasoning. We note that the benchmark exhibits answer-level imbalance, where a small subset of answers dominates the distribution within certain question categories. This effect is amplified by tool–tissue interaction labels (e.g., *Idle*, *Tissue Manipulation*), which is highlighted by the large gap between oracle and non-oracle settings in terms of Recall and F1.

**Language models.** We compare multiple frozen LLMs to examine how the proposed pipeline behaves across language models of different sizes and origins, serving as a robustness analysis under identical specialist-derived inputs. Table 2 compares frozen LLMs across three input settings: specialist predictions (FASL+INT), oracle annotations, and an image-only condition. With specialist inputs, GPT-5 achieves the strongest performance (77.4% accuracy, 47.6% F1), with GPT-5-mini and GPT-5-nano close behind. DeepSeek-chat and DeepSeek-reasoner perform competitively but remain several points lower in F1, while GPT-4o/4o-mini and the Qwen-VL models lag further. Under oracle inputs, all models improve substantially: GPT-5-mini reaches the highest scores (93.0% accuracy, 95.3% F1), with GPT-5 and the stronger Qwen-VL variants close behind. Overall, the relatively small performance gap observed across language models when structured specialist outputs are available suggests that the pipeline operates robustly across LLM choices, rather than being dependent on a specific model.

**Perception quality vs. VQA performance.**

To further analyze the contribution of specialist models, we relate task-specific perception metrics to end-to-end VQA accuracy. Figure 3 shows how VQA performance varies as a function of segmentation quality (mIoU) and interaction prediction accuracy, evaluated on the same data split used for VQA. Improvements in segmentation primarily benefit spatial questions, while interaction prediction quality most strongly affects interaction-related questions, consistent with Table 2, with consistent trends observed across different LLMs. Notably, VQA accuracy increases more steeply with interaction prediction accuracy, starting near zero when interactions are incorrect, reflecting that incorrect interaction recognition typically precludes answering interaction-specific questions. In contrast, segmentation-driven performance degrades more gracefully, as partially accurate or slightly misaligned masks can still support correct answers for a subset of spatial queries.

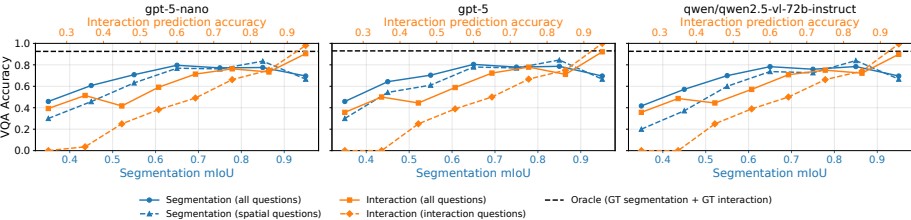

Figure 3: Relationship between specialist model performance and end-to-end VQA accuracy across different LLMs. Each subplot corresponds to a frozen LLM and reports VQA accuracy as a function of segmentation quality (mIoU, bottom axis) and interaction prediction accuracy (top axis), further broken down by question type. The dashed horizontal line denotes oracle performance obtained using ground-truth segmentation and interaction labels.

**Image-only reasoning.** The image-only condition in Table 2 evaluates each LLM using only the raw endoscopic frame. This isolates the contribution of our modular pipeline: the same LLM is evaluated once with specialist-derived facts and once with no structured perception. Performance drops sharply across all models (e.g., GPT-5: 37.2% accuracy,

Table 2: Comparison of LLMs on EndoVis-18-VQA under three input settings: (i) **FASL, INT** uses predictions from both specialists, (ii) **Oracle** uses ground-truth masks and interaction labels, and (iii) **Only Images** provides the raw frame with no specialist input. Accuracy, Recall, and F1 are reported in percent. Cost and Latency columns show the average LLM cost (in cents) and LLM inference latency (in seconds) per sample. Bold values indicate the best result within each setting (maximum for performance metrics and minimum for efficiency metrics).

| | FASL, INT | | | | | Oracle | | | | | Only Images | | |
|---|---|---|---|---|---|---|---|---|---|---|---|---|---|
| | Acc | Recall | F1 | Cost | Lat. | Acc | Recall | F1 | Cost | Lat. | Acc | Recall | F1 |
| deepseek-chat | 74.29 | 45.04 | 45.51 | 0.017 | 1.87 | 92.56 | 94.74 | 93.67 | 0.015 | 1.88 | – | – | – |
| deepseek-reasoner | 75.91 | 40.72 | 41.27 | 0.047 | 29.37 | 92.96 | 89.04 | 88.89 | 0.034 | 20.04 | – | – | – |
| gpt-4o | 74.43 | 39.85 | 40.64 | 0.306 | 0.75 | 91.55 | 71.35 | 72.36 | 0.265 | 0.84 | – | – | – |
| gpt-4o-mini | 72.99 | 35.16 | 36.05 | 0.037 | 0.76 | 89.96 | 82.42 | 83.04 | 0.032 | 0.80 | – | – | – |
| gpt-5 | **77.39** | **47.06** | **47.58** | 0.190 | 9.45 | 92.99 | 95.00 | 94.81 | 0.115 | 3.04 | **37.23** | **29.79** | **28.54** |
| gpt-5-mini | 76.56 | 46.56 | 47.23 | 0.029 | 3.14 | **93.03** | **95.02** | **95.28** | 0.021 | 2.61 | – | – | – |
| gpt-5-nano | 76.60 | 43.73 | 44.47 | 0.012 | 3.05 | 92.52 | 88.90 | 86.08 | 0.009 | 2.63 | 34.38 | 24.96 | 21.36 |
| qwen2.5-vl-32b | 71.98 | 18.18 | 18.27 | **0.003** | 1.00 | 90.47 | 46.71 | 46.03 | **0.003** | 0.93 | 27.23 | 5.93 | 3.68 |
| qwen2.5-vl-72b | 75.62 | 36.29 | 36.84 | 0.010 | **0.72** | 92.60 | 88.86 | 89.06 | 0.008 | **0.65** | 36.04 | 15.44 | 12.52 |
| qwen3-vl-8b | 72.44 | 26.41 | 27.06 | 0.005 | 1.40 | 90.75 | 63.84 | 64.10 | 0.005 | 1.40 | 26.25 | 9.74 | 7.65 |

Latency is measured per query and reflects end-to-end API response time, including network overhead and variability due to server load.

28.5% F1), showing that even strong LLMs struggle to infer clinically precise spatial or semantic cues from pixels alone. The consistent gap between image-only and specialist-assisted performance highlights the importance of the modular abstraction stage in enabling frozen LLMs to operate effectively in surgical settings without multimodal fine-tuning.

**Baselines.** Table 3 summarizes performance on EndoVis-18-VQA. Early transformer-based approaches such as VisualBERT and its residual variant (VisualBERT RM) achieved accuracies around 61–62%, with limited recall and F1, reflecting the difficulty of learning surgical semantics from limited VQA supervision. More recent efforts sought to improve reasoning by augmenting vision features. SurgicalGPT (LV-GPT) introduced a GPT-based architecture with vision token embeddings and careful token sequencing, reaching 66–68% accuracy. Surgical-LVLM further adapted a large vision-language model with Visual Perception LoRA modules, attaining close to 70% accuracy. The most recent large-scale effort, SurgVLM, leveraged instruction tuning on multimodal surgical data and achieved 75.0% accuracy (Zeng et al., 2025).

In contrast, our proposed modular pipeline, which avoids multimodal fine-tuning altogether, surpasses these baselines. With GPT-5-nano, Surg-SAGE achieves 76.6% accuracy, while GPT-5 attains 77.4% accuracy with the highest F1 score (47.6%). These results show that structured, specialist-driven scene abstraction combined with frozen LLMs can outperform both earlier transformer models and recent foundation-scale VLMs.

**Efficiency and computational cost.** Beyond predictive performance, we analyze computational efficiency in terms of model scale, inference latency, and cost. Table 3 reports parameter counts where available. Early surgical VQA models such as VisualBERT RM (159.1M) and LV-GPT (RN18) (175.0M) are relatively lightweight, whereas more recent approaches scale substantially, including Surgical-LVLM (7B) and SurgVLM-72B (72B). Surg-SAGE spans a comparable range: the Qwen2.5-VL-72B instantiation matches

Table 3: Performance comparison between prior surgical VQA approaches and Surg-SAGE on EndoVis-18-VQA. ‡ denotes variants belonging to Surg-SAGE.

| Model | Params | Acc | Recall | F1 |
|---|---|---|---|---|
| VisualBert (Li et al., 2019) | 184.2M | 61.43 | 42.82 | 37.45 |
| VisualBert RM (Seenivasan et al., 2022a) | 159.1M | 61.90 | 40.79 | 35.83 |
| LV-GPT (RN18) (Seenivasan et al., 2023) | 175.0M | 68.11 | 46.49 | 46.49 |
| LV-GPT (Swin) (Seenivasan et al., 2023) | 191.0M | 66.13 | 44.60 | 45.37 |
| Surgical-LVLM (Wang et al., 2024a) | 7.0B | 69.47 | – | 33.25 |
| SurgVLM-72B Lora-tuning (Zeng et al., 2025) | 72.0B | 75.02 | – | – |
| **Surg-SAGE (qwen3-vl-8b)‡** | 8.1B | 72.44 | 26.41 | 27.06 |
| **Surg-SAGE (gpt-5-nano)‡** | – | 76.60 | 43.73 | 44.47 |
| **Surg-SAGE (gpt-5)‡** | – | **77.39** | **47.06** | **47.58** |

SurgVLM-72B in scale while achieving slightly higher accuracy (75.62% vs. 75.02%), and smaller variants such as Qwen3-VL-8B attain competitive performance at similar model sizes. For proprietary API-based LLMs (e.g., OpenAI), parameter counts are not publicly disclosed and are therefore omitted.

Table 2 further reports average LLM inference cost and latency per sample. Results show that strong performance is achievable with relatively low-cost models: GPT-5-nano outperforms all prior surgical VQA baselines at an average cost of 0.012 cents per sample, while Qwen3-VL-8B achieves competitive accuracy at 0.005 cents per sample, indicating that expensive foundation models are not a prerequisite within the proposed framework. LLM inference latency varies across models, ranging from sub-second to several seconds depending on model size and API conditions (e.g., 3.05 s for GPT-5-nano and 9.45 s for GPT-5). Specialist inference time is constant across configurations (0.78 s), with an additional 0.19 s for heuristic abstraction, yielding end-to-end latencies of 4.02 s (GPT-5-nano) and 10.42 s (GPT-5). Earlier baselines exhibit lower latency due to smaller model size (e.g., VisualBERT RM: ∼58 ms, LV-GPT (Swin): ∼80 ms), and the absence of network overhead and API-related variability (e.g., server load or request queuing), while latency figures for larger, more recent models (e.g., SurgVLM) are not publicly available at the time of writing. All reported timings are measured under identical conditions as described in Section 4.

**Discussion.** Unlike prior approaches that depend on multimodal pretraining or direct VQA-style supervision, Surg-SAGE achieves superior performance by decoupling perception from reasoning. Models such as VisualBERT RM and Surgical-GPT rely on VQA-style training, whereas SurgVLM leverages large-scale multimodal pretraining. In contrast, Surg-SAGE requires only task-specific training of vision specialists, offering a more data-efficient and generalizable paradigm. The textual intermediate representation provides interpretability, and the modular design enables seamless incorporation of additional experts for new skills or procedures, allowing the pipeline to adapt to new datasets through strongly supervised specialist training rather than expensive multimodal data collection or fine-tuning of the language model.

Importantly, the proposed modular abstraction strategy (Figure 1) is benchmark-agnostic and can be applied to other surgical datasets by swapping or extending specialist models,

without retraining the language component. For example, temporal experts such as phase recognition or action triplet models can be incorporated to summarize procedural context over time using the same abstraction interface. Furthermore, an alternative abstraction strategy is to overlay specialist outputs (e.g., segmentation masks and labels) directly on the image and prompt a VLM. We observed that, in practice, this approach is sensitive to noise in predicted masks, visually cluttered in complex scenes, and incurs higher visual token costs compared to the proposed structured textual abstraction.

**Clinical feasibility and deployment scope.** In its current form, the pipeline is not intended for real-time intra-operative use, as end-to-end latency is dominated by LLM inference (API response time), with additional contributions from specialist inference (0.78 s) and abstraction (0.19 s), resulting in $\sim$4.02 s/query with GPT-5-nano and $\sim$10.42 s/query with GPT-5. This latency profile is nevertheless well suited to offline and retrospective workflows where seconds-level response times are acceptable, including postoperative review, surgical education, quality improvement, and semi-automated annotation. Specialist outputs and their textual abstractions can be pre-computed and cached per frame, such that additional queries incur only the LLM stage, enabling faster interactive analysis during video review. As for operating cost, the LLM component is relatively inexpensive (e.g., 0.012 cents/query for GPT-5-nano), which supports large-scale retrospective analysis and dataset generation from existing surgical video archives. Taken together, these properties position the proposed system as an auditable reasoning and data-generation backbone that complements real-time surgical AI, providing transparent intermediate facts and scalable supervision for training or distilling faster, lower-latency models.

**Limitations.** The current evaluation focuses on vision-centric, frame-level queries in EndoVis-18-VQA, which does not capture knowledge-intensive or temporally extended reasoning. As such, the benchmark primarily evaluates spatial and functional scene understanding rather than long-horizon temporal aggregation or external clinical knowledge. While the pipeline is expected to generalize with appropriate specialists, broader validation across additional surgical datasets and procedures remains an important direction for future work. The modular design, while flexible, introduces additional computational overhead at inference time since multiple experts must be executed before language inference. Future work will study lightweight expert-routing mechanisms that activate only the relevant specialists. Extending the pipeline to temporal experts for video understanding, evaluating on larger and more diverse surgical datasets, and assessing performance on temporally complex queries represent natural next steps.

## 6. Conclusion

In conclusion, abstraction-driven modularity emerges as a viable alternative to end-to-end multimodal training for surgical vision-language understanding. By separating perception from reasoning and channeling expert predictions into structured textual facts, the proposed Surg-SAGE approach achieves state-of-the-art performance on the Endovis-18-VQA surgical benchmark while requiring no multimodal pretraining. These findings highlight both the promise and the trade-offs of modular abstraction, underscoring its potential as a practical foundation for future extensions in surgical intelligence.

## Acknowledgments

Research reported in this publication was supported by the Qatar Research Development and Innovation Council (QRDI) grant number ARG01-0522-230266. Disclaimer: The content is solely the responsibility of the authors and does not necessarily represent the official views of Qatar Research Development and Innovation Council.

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

## Appendix A. Heuristics for Textual Abstraction

This appendix provides the full technical details of the deterministic heuristics used to convert segmentation masks and interaction predictions into the textual fact sets consumed by the LLM. These procedures correspond directly to the high-level description in Section 3.2.1.

### A.1. Segmentation Heuristics

Given a semantic segmentation mask

$$S \in \{0, \ldots, C\}^{H \times W},$$

where each pixel encodes one of $C$ semantic classes, the segmentation heuristics module $H_{\text{seg}}$ produces a fact set $\mathcal{T}_{\text{seg}}$ summarizing object presence, layout, and spatial relationships. As in the main paper, three families of statements are emitted: *absolute location*, *pairwise spatial relations*, and *graded proximity*.

**Absolute location.** The goal of this step is to map each object instance into a coarse symbolic location such as "top-left" or "center." Formally, let

$$O_k = \{(i,j) \mid S_{i,j} = k\}$$

denote the set of pixels belonging to class $k$. We consider a collection of spatial partitions $\mathcal{P}$, each of which defines regions of the image grid. Importantly, these partitions are applied hierarchically where *fine-grained regions* such as quadrants ("top-left," "bottom-right," etc.) are tested first, and only if no assignment is found do we fall back to *coarser partitions* such as halves ("left," "right," "top," "bottom").

For each region $R \in \mathcal{P}$, we measure the overlap with the object pixels:

$$\frac{|O_k \cap R|}{|O_k|}.$$

If at least $\tau = 0.75$ (75%) of the object lies within $R$, the object is assigned to that region. If no assignment succeeds at the class level, we decompose $O_k$ into connected components $\{O_{k,r}\}_{r=1}^m$ and repeat the test per component, yielding a (possibly multi-tag) set of locations for $k$. This design gives a single stable tag when the majority of the object lies in a single region, yet captures multi-part layouts when the class appears in multiple regions. Figure 2 (a) shows an illustration of this hierarchical procedure.

**Pairwise spatial relations.** While absolute positions capture coarse symbolic regions, they do not express relational cues when objects occupy the same area and have irregular shapes. To address this, we define pairwise spatial relations based on dilation-based adjacency.

For each object $k$, we dilate its mask $S_k$ by $\rho$ pixels,

$$S_k^{(\rho)} = \text{dilate}(S_k, \rho),$$

which intuitively thickens the object's boundary by $\rho$ pixels in every direction. Two objects $k$ and $\ell$ are considered adjacent if the dilated support of one intersects the original mask of the other:

$$S_k^{(\rho)} \cap S_\ell \neq \emptyset.$$

Once adjacency is established, we use the centroids $\mathbf{c}_k$ and $\mathbf{c}_\ell$ to infer the dominant orientation of $\ell$ with respect to $k$, yielding interpretable statements such as "the forceps is to the left of the needle." Importantly, this procedure complements absolute locations: when two objects are well separated across regions, their relative order is already captured (e.g., an object in the top-right is naturally to the right of one in the top-left). Pairwise relations are thus only triggered in more complex cases where objects lie within the same area and require finer geometric reasoning. Figure 2 (b) illustrates this process, showing for example how the dilated mask of the monopolar curved scissors intersects with the kidney parenchyma but not with the adjacent prograsp forceps.

***Graded proximity.*** Whereas absolute positions provide coarse layout and pairwise adjacency captures direct contact, graded proximity introduces a softer notion of distance between instruments and tissues. For each instrument mask $S_k$, we generate a sequence of dilations $\{S_k^{(r_m)}\}_{m=1}^L$ with increasing radii $r_m$. For a tissue mask $S_\ell$, we then record the smallest index $m$ such that

$$S_k^{(r_m)} \cap S_\ell \neq \emptyset.$$

This index is mapped to a discrete label (e.g., "touching," "very close to," "close to," or "far from"). In this way, graded proximity enriches the spatial description by quantifying not only whether an instrument is in contact with tissue, but also the degree of closeness, offering a more nuanced account of their spatial relationship.

### A.2. Interaction Heuristics

Unlike segmentation masks that require geometric reasoning, the output of the interaction model is already provided in a predefined categorical format, indicating functional relations such as "grasping," "cutting,"or "idle." The heuristics module $H_{\text{int}}$ converts these categorical predictions into declarative textual facts using simple templates. For example:

prograsp forceps: "Tissue Manipulation"
monopolar curved scissors: "Idle"

This yields a fact set $\mathcal{T}_{\text{int}}$ that directly summarizes the functional roles of instruments in the current frame. Combined with the spatial cues from segmentation, these interaction statements provide complementary information, enabling the LLM to reason not only about *where* objects are but also about *how* they are used.

## Appendix B. Sensitivity Analysis of Heuristic Parameters

To assess the robustness of the proposed textual abstraction heuristics, we conduct a sensitivity analysis over key geometric parameters used in the segmentation-based rules, including dilation radii and proximity expansion thresholds. These parameters control adjacency detection and graded proximity assignment

Table 4 reports end-to-end VQA performance under three representative heuristic configurations evaluated on the same split used in the main experiments. Across a wide range of dilation and expansion settings, performance remains stable, with only minor variations in Accuracy, Recall, and F1. This behavior indicates that the gains of the proposed pipeline stem from the abstraction mechanism itself, i.e., the structured conversion of perception outputs into symbolic facts, rather than from careful tuning of individual geometric thresholds, demonstrating robustness of the text synthesis step to reasonable parameter choices.

## Appendix C. Additional Qualitative Examples

### C.1. Hybrid Segmentation Illustration (SAM+FASL)

Figure 4 provides a qualitative illustration of the hybrid segmentation procedure described in Section 3.1.1.

Table 4: Effect of dilation and expansion settings on segmentation performance.

|  | Dilation Levels | Expansion (px) | Accuracy | Recall | F1 |
|---|---|---|---|---|---|
| **Config-1** | 5, 10, 30 | 10 | 0.7678 | 0.4670 | 0.4745 |
| **Config-3** | 10, 15, 40 | 15 | 0.7627 | 0.4094 | 0.4160 |
| **Config-2** | 15, 20, 50 | 30 | 0.7624 | 0.4351 | 0.4416 |

The region-proposal component generates a set of candidate masks, and each region is assigned a semantic class via majority voting over the predictions of the specialist segmentation model. This visualization highlights how coarse region proposals can be combined with domain-aware semantics to produce stable, coherent segmentation outputs.

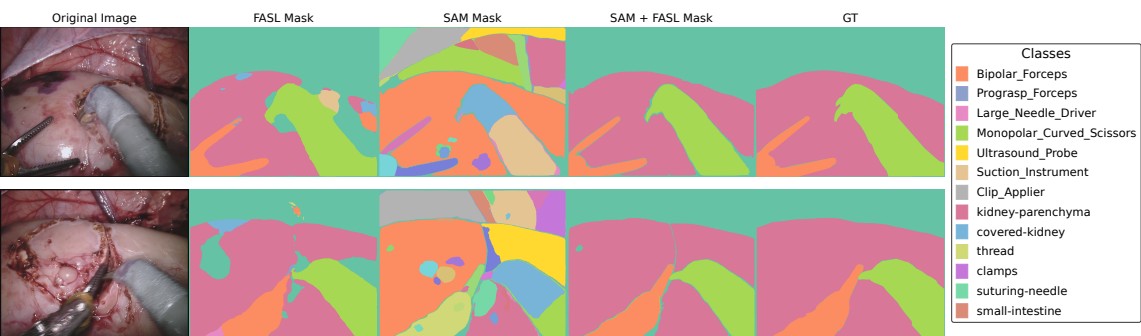

Figure 4: Example of SAM+FASL segmentation. SAM provides mask proposals, while FASL supplies domain-specific semantics. GT denotes ground truth.

