# OpenReview forum: "Decoupling Vision and Reasoning: A Data-Efficient Pipeline for Surgical VQA"
_MIDL.io/2026/Conference — MIDL 2026 Poster_

### Official Review · Reviewer_K57y · 2026-01-02

**Confidence:** 4
**Preliminary Rating:** 2
**Final Rating:** 3

**Summary:**

The paper proposes a modular surgical VQA pipeline that decouples visual perception from language-based inference by first applying task-specific specialist models (e.g., segmentation and tool–tissue interaction) and then synthesizing their outputs into textual descriptions consumed by a frozen LLM.

Experiments on the EndoVis-18-VQA benchmark show that this text abstraction enables competitive or superior accuracy compared to end-to-end multimodal VQA baselines, without requiring multimodal training of the language model.

The work shows that carefully designed textual abstractions can effectively bridge specialist vision models and LLMs, offering a data-efficient alternative to fully trained surgical vision–language models.

**Strengths:**

- The paper presents a clear modular framework that separates visual perception from reasoning by converting specialist model outputs into texts for LLM-based inference. This design is valuable because it directly addresses the data scarcity problem in surgical VQA, avoiding expensive multimodal training while maintaining interpretability and flexibility.

- The experimental evaluation is thorough, including ablations over different specialist combinations, comparisons across multiple LLMs, and experiments where specialist prediction are replaced with ground-truth labels, which clarify where performance bottlenecks lie.

- The method is easy to extend by swapping or adding specialists, which increases its practical relevance for the community as new surgical perception models become available.

**Weaknesses:**

- The algorithmic contribution of the paper is limited. The core idea is to run specialist perception model, convert their outputs into structured texts (w/ deterministic rules), and feed the question along with the texts to a frozen LLM. Though this is different from end-to-end trained surgical VQA models or models with scene graphs, the new methodological component is mainly the text synthesis (i.e. templating heuristics such as location, pairwise relations, proximity), rather than a new learning algorithm. Therefore, as a methods paper, careful validation of robustness, generalization, and cost are expected.
- Table 2 shows that the performance with Oracle is dramatically better than that with specialist model predictions, which indicates that either the accuracy of FASL/SAM2 and INT are poor or the proposed framework is sensitive to the noise in specialist model predictions. The manuscript could benefit from 1) report the performance of the specialist models; 2) an in-depth analysis of the impact of model prediction error on the performance of overall framework, both quantitatively and qualitatively. For example, the authors can purposely induce error into specialist predictions and plot the relationship between the specialist performance and the overall performance. The authors can also show which perception failures dominate VQA errors (e.g., mis-segmentation vs. missed tool presence vs. incorrect interaction state) and how these propagate through the heuristics into incorrect facts.
- Specialist model training/annotation cost are underreported relative to the paper’s “data-efficient” motivation. A concrete accounting of annotation volume and cost for training specialists (frames labeled, label types, estimated expert time), and how that compares to collecting VQA annotations for baseline approaches should be beneficial.
- The text abstraction step uses hard-coded rules ( hierarchical spatial bins, adjacency from dilation, and discretized proximity labels), which could introduce systematic brittleness. The paper could benefit from a sensitivity analysis over key heuristic parameters and a demonstration that gains are not contingent on a narrow set of hand-tuned rules.
- The paper argues clinical relevance, but it does not define what performance would be clinically usable, nor does it analyze whether remaining errors are low-stakes or high-stakes. A more clinically grounded error analysis would better justify the practical value of the method.
- Since the method’s key claim is that textual abstraction makes reasoning data-efficient, an important missing baseline is: overlaying specialist outputs (masks/labels) on images and prompting a vision–language model. This would directly test whether the benefit comes from the text interface, the specialists themselves, or the LLM’s language reasoning.

**Detailed Comments:**

As mentioned in Weaknesses:

- Report standard task-specific metrics for the specialist models (e.g. Dice/mIoU for segmentation, per-class accuracy or F1 for interaction) on the same data split used for VQA evaluation.
- Add a brief sensitivity study on key heuristic parameters (e.g. spatial partition thresholds, proximity bins) to demonstrate robustness of the text synthesis step.
- Explicitly discuss computational cost and latency of running multiple specialists at inference time, and comment on feasibility for clinical use.

**Justification Of Final Rating:**

The revision substantially improves the paper’s quality. I appreciate the detailed analysis of how the vision model’s performance affects VQA performance, the sensitivity study of key heuristic parameters, as well as the discussion of cost, speed, and clinical usability. That said, I believe the proposed method relies on manual context engineering and may not generalize easily to other types of surgical videos. As a result, its overall impact may be limited. I will raise my rating to 3.

**Justification Of The Preliminary Rating:**

While the paper presents a well-engineered modular pipeline for surgical VQA, its algorithmic novelty is limited, with the main contribution lying in deterministic text synthesis from specialist model outputs rather than in a new learning approach. Given its position as a methods paper, the evaluation does not yet meet the level of rigor needed to fully substantiate its advantages in robustness, data efficiency, and clinical relevance. In particular, the lack of reported specialist model performance, insufficient analysis of error propagation from perception to LLM inference, and the absent discussion of annotation cost needed to train the specialist models weaken the significance of the proposed method. Moreover, the large performance gap between oracle and specialist settings suggests that the framework may be highly sensitive to perception noise, yet this sensitivity is not systematically analyzed. Overall, the work shows clear potential and practical interest, but would benefit from additional validation, robustness analysis, cost analysis, and clinical-oriented evaluation.

**Questions To Address In The Rebuttal:**

As mentioned in Weaknesses:

- Since the algorithmic innovation itself is limited, the paper should benefit from an in-depth analysis of robustness (e.g. the impact of model prediction error on oeverall performance; what types of errors are most detrimental) and cost (e.g. annotation cost for training specialist models).
- Performance analysis from a clinical perspective.
- Comparison with overlaying specialist outputs (masks/labels) on images and prompting a VLM.

---

> ### Author Response · Authors · 2026-01-24
>
> 1.	C1: Since the algorithmic innovation itself is limited, the paper should benefit from an in-depth analysis of robustness (e.g. the impact of model prediction error on overall performance; what types of errors are most detrimental) and cost (e.g. annotation cost for training specialist models).
> C1.1: Report standard task-specific metrics for the specialist models (e.g. Dice/mIoU for segmentation, per-class accuracy or F1 for interaction) on the same data split used for VQA evaluation.
>
> We thank the reviewer for this helpful suggestion. In the revised manuscript, we added an additional analysis relating task-specific specialist performance to end-to-end VQA accuracy. Specifically, we report how segmentation quality (mIoU) and interaction prediction accuracy affect VQA performance on the same data split used for evaluation, with results summarized in a new figure in the Results and Discussion. This analysis clarifies the contribution of each specialist and shows how perception quality directly governs downstream VQA performance. We refer the reviewer to the revised manuscript with highlighted updates.
>
> C1.2: Add a brief sensitivity study on key heuristic parameters (e.g. spatial partition thresholds, proximity bins) to demonstrate robustness of the text synthesis step.
>
> We thank the reviewer for this suggestion. To address robustness of the text synthesis step, we have added a sensitivity study in the appendix of the revised manuscript analyzing the impact of key heuristic parameters (spatial dilation levels and expansion thresholds) on end-to-end VQA performance. The results, reported in a new table and accompanying paragraph, show that performance remains stable across a broad range of parameter settings, indicating that the proposed abstraction mechanism is robust and not dependent on fine-tuned thresholds. We refer the reviewer to the revised manuscript with highlighted updates.
>
>
>
> 2.	C2: Performance analysis from a clinical perspective.
> C2.1: Explicitly discuss computational cost and latency of running multiple specialists at inference time, and comment on feasibility for clinical use.
>
> We thank the reviewer for this clinically motivated comment. In response, we explicitly analyze the computational cost and inference latency incurred by running multiple specialist models and the LLM stage, reporting per-sample latency and cost in the revised manuscript. Specifically, we extend Table 2 to include average LLM inference latency and cost per sample and add a dedicated efficiency paragraph in the Results and Discussion analyzing computational cost, latency, and model scale in comparison to existing baselines. A discussion of practical considerations for clinical use is also included in the Discussion section. We refer the reviewer to the revised manuscript with highlighted updates.
>
>
>
> 3.	C3: Comparison with overlaying specialist outputs (masks/labels) on images and prompting a VLM.
>
> We thank the reviewer for this suggestion. Overlaying specialist outputs (e.g., segmentation masks and class labels) directly on images and prompting a vision–language model is indeed a possible alternative abstraction. In practice, however, we found this approach to be sensitive to noise in predicted masks and visually cluttered in frames containing multiple instruments and anatomical structures, which can hinder effective perception/reasoning. Even under oracle settings, such visual abstractions did not yield competitive performance (lower than 50% accuracy), indicating that reasoning directly over dense visual overlays remains challenging in this domain. In addition, this strategy incurs higher visual token costs and relies heavily on visual reasoning, whereas LLMs are generally more reliable when operating in the language modality.
> In the revised manuscript, we now explicitly mention this alternative abstraction strategy and discuss its limitations in the Discussion section, motivating our choice of structured textual abstraction. We refer the reviewer to the revised manuscript with highlighted updates. To further illustrate this point, we include a qualitative example in the rebuttal showing how mask overlays can become visually dense and ambiguous in realistic surgical scenes (In the supporting material of the rebuttal).

---

### Official Review · Reviewer_Bsp4 · 2026-01-09

**Confidence:** 3
**Preliminary Rating:** 3
**Final Rating:** 4

**Summary:**

This paper introduces a modular Surgical VQA pipeline that decouples perception from reasoning to improve data efficiency. The authors utilize frozen "specialist" models (segmentation and interaction) to extract visual features, which are converted into structured text heuristics for a frozen LLM (e.g., GPT-5). Evaluated on the EndoVis-18-VQA benchmark, the method achieves competitive performance (77.4% accuracy), outperforming end-to-end baselines like SurgVLM while offering transparent intermediate representations.

**Strengths:**

- Data Efficiency: The approach effectively sidesteps the scarcity of surgical multimodal data by leveraging existing specialist models and off-the-shelf LLMs, avoiding the need for expensive pre-training.
- Interpretability: The intermediate textual representation (e.g., explicit tool states) serves as a transparent bottleneck, allowing clinicians to verify the exact visual facts used for reasoning.
- Performance: The method demonstrates strong results on EndoVis-18, surpassing state-of-the-art models (SurgVLM, SurgicalGPT), validating the potential of symbolic approaches in closed domains.

**Weaknesses:**

- Limited Validation Scope: The evaluation is restricted to a single dataset (EndoVis-18). Without testing on a second domain, the claim of "generalizability" is unproven, particularly regarding the transferability of the spatial heuristics.

- Brittleness of Heuristics: The system relies on manually defined rules (e.g., spatial thresholds). This restricts the VQA capabilities to a closed set of questions; unlike end-to-end models, it cannot answer queries about visual attributes not explicitly coded (e.g., smoke, bleeding).

- Missing Computational Analysis: There is no analysis of inference latency or cost. Running multiple specialist models followed by a commercial LLM API call is likely significantly slower and more expensive than efficient end-to-end baselines.

- Incremental Novelty: The concept of "Modular VQA" is well-established in general computer vision. The novelty is primarily the application to surgery rather than a methodological innovation.

**Detailed Comments:**

- Metrics: Please add a column to Table 3 comparing the inference time (FPS) and estimated cost of your method versus the baselines.
- Performance: An additional comparison to fine-tuned models would be interesting to better gauge the trade-off between the "frozen" approach and domain adaptation.
- Dataset: You mention "To avoid data leakage, videos 1, 5, and 16 are held out..."; please explain this choice. Is this a standard split in the community? Furthermore, please discuss additional datasets your approach could be tested on.
- Figure Captions: The figure captions (e.g., Figure 1 and 2) are currently quite brief. Please enhance them to be more self-contained, clearly describing the inputs, outputs, the flow, etc.

**Justification Of Final Rating:**

The authors substantially improved the manuscript during the rebuttal by adding the requested computational cost/latency analysis and clarifying the data split strategy. While I remain unconvinced that the method is conceptually distinct from prior literature, and the reliance on heuristics limits generalization, the practical engineering value and data efficiency of the pipeline are well-supported. I have raised my score to 4. (Minor note: for the final camera-ready version, please consider moving the caption of figure 2 below the graphic)

**Justification Of The Preliminary Rating:**

The paper proposes a practical, data-efficient solution with good benchmark results. However, the reliance on manual heuristics limits true generalizability, and the lack of cost/latency analysis hides potential deployment barriers. The validation on a single dataset further restricts the impact; the authors are asked to clarify runtime costs and robustness to new domains before a higher rating can be considered.

**Questions To Address In The Rebuttal:**

**Inference Latency & Cost:** What is the end-to-end latency (FPS) and cost per minute of analysis? This is critical to determine if the "frozen" advantage is practical for clinical use.

**Generalization:** Have you tested these specific heuristics on a different surgical procedure? Do the spatial partition rules hold up when the anatomical view changes?

**Out-of-Scope Queries:** How does the system handle questions regarding visual attributes not captured by the specialists (e.g., "Is the view obscured by smoke?")?

---

> ### Author Response · Authors · 2026-01-24
>
> 1.	C1: Metrics: Please add a column to Table 3 comparing the inference time (FPS) and estimated cost of your method versus the baselines.
>
> We thank the reviewer for this suggestion. Following the reviewer’s request to report inference efficiency, we considered FPS as a potential metric. However, since in our setting a single image can be associated with multiple questions, a per-frame FPS measure can be ambiguous. We therefore report per-sample inference latency instead, which more directly reflects the cost of answering an individual VQA query. In the revised manuscript, we extend Table 2 to include average LLM inference latency and cost per sample and add a dedicated efficiency paragraph in the Results and Discussion analyzing computational cost, end-to-end latency, and model scale in comparison to existing baselines. We thank the reviewer for raising this point, which helped improve the clarity and completeness of the manuscript, and refer them to the revised manuscript with highlighted updates.
>
>
> 2.	C2: Performance: An additional comparison to fine-tuned models …
>
> We thank the reviewer for their thoughtful comment. Indeed, comparing against fine-tuned models is essential gauge the trade-off between the frozen approach and domain adaptation and to emphasize the superiority of the proposed approach to end-to-end training. The baselines included in Table 3 are all trained in an end-to-end manner on the dataset at hand. For example, SurgVLM, which achieves state-of-the-art performance prior to our work, is a Qwen2.5-VL model trained on a large collection of datasets including Endovis-18-VQA. To avoid confusion, in the revised manuscript, we highlight that the baselines are end-to-end trained as opposed to our modular pipeline in the results section.
>
>
> 3.	C3: Dataset: You mention “To avoid data leakage…”
>
> We thank the reviewer for pointing out this ambiguity. The reason we explicitly state the held-out videos is that the splits used in EndoVis-18-VQA (for VQA evaluation) are defined differently from the splits typically used when training perception specialists on EndoVis-18 segmentation annotations. As a result, there can be overlap between frames/videos used to train a segmentation specialist and the test split of EndoVis-18-VQA. If a specialist is trained on data that overlaps with the VQA test set, this would constitute data leakage and could lead to unfair comparisons with VQA baselines.
> To avoid this, we retrain all specialist models using only videos that do not appear in the EndoVis-18-VQA test set, holding out videos 1, 5, and 16 consistently across all specialists. This ensures that no visual content from the VQA test split is seen during specialist training. We have now clarified this rationale in the experimental setup section of the camera-ready version.
>
>
> 4.	C4: Figure Captions
>
> The authors thank the reviewer for their comment regarding improving the clarity of the manuscript. In the camera-ready version, the captions of Figures 1 and 2 have been extended to be more self-contained and to clearly describe the inputs, outputs, and information flow of the proposed pipeline.
>
>
> 5.	Generalization
>
> We thank the reviewer for this question. We evaluate our approach on EndoVis-18-VQA as it is a widely used benchmark in surgical VQA and captures core frame-wise capabilities such as spatial reasoning (via segmentation) and functional reasoning (via instrument–tissue interaction). Importantly, the proposed heuristics are task-driven rather than dataset- or procedure-specific: the spatial partitioning, proximity, and relational rules operate on normalized image geometry and relative object layouts, and are therefore agnostic to the specific surgical procedure. Our objective in this work is to first establish the effectiveness of abstraction-driven modularity in a controlled setting before extending to additional procedures. Accordingly, we have strengthened the discussion and limitations sections of the revised manuscript to explicitly outline this extensibility and leave cross-procedure evaluation for future work.
>
>
> 6.	Out-of-Scope Queries
>
> We thank the reviewer for raising this point. Similar to end-to-end trained VQA and vision–language models, the proposed system can only answer questions for which corresponding perceptual capabilities are available. Queries involving visual attributes not modeled by the current specialists (e.g., smoke or occlusion) are therefore out of scope for the present instantiation. The modular design of our framework allows such capabilities to be incorporated by adding new specialist models, and this generalizability is now emphasized in the discussion section of the revised manuscript. In addition, in the camera-ready version, we will include a brief qualitative analysis illustrating the limits of the current pipeline on out-of-scope queries, alongside a comparison with baseline approaches.

---

### Official Review · Reviewer_fkCd · 2026-01-13

**Confidence:** 5
**Preliminary Rating:** 3
**Final Rating:** 4

**Summary:**

The manuscript proposes a modular pipeline decoupling vision extraction from reasoning, using specialist models and heuristics for text conversion, enabling LLM reasoning without multimodal training. Experiments are conducted on EndoVis-18-VQA with various specialist modelsand LLMs, comparing against surgical VQA baselines.

**Strengths:**

1. The manuscript decouples vision extraction from reasoning, targeting the scarcity of surgical multimodal datasets.
2. Integrates specialist vision models (segmentation/interaction) for complementary cues, combining with LLMs to balance performance and interpretability, with strong practical value.
3. Requires no multimodal pretraining, avoiding overfitting and data reliance, lowering clinical application barriers, meaningful for real-world adoption.

**Weaknesses:**

1. The proposed framework shows promise for solid VQA performance at low cost. However, the experimental analysis could be strengthened. Metrics like F1 have limitations in reflecting VQA performance directly—adding indicators such as BLEU-4 would make the assessment more comprehensive.
2. It would also help if the authors clarify which design choices allow the pipeline to work without multimodal training?
3. Since no LLMs were fine-tuned, it would be valuable to explain the significance of comparing different LLMs in experiments.

**Detailed Comments:**

Please check the Weaknesses part.

**Justification Of Final Rating:**

Thanks to the author’s clarification, some of my concerns have been resolved. However, in the experiment section, I recommend the author introduce more metrics to evaluate its performance to avoid reducing the VQA problem to a classification problem.  I therefore give a final rating of 4.

**Justification Of The Preliminary Rating:**

The research addressing surgical VQA’s multimodal dataset scarcity via a decoupled modular pipeline is valuable. However, the manuscript inadequately explains the pipeline’s working mechanism—both its core design rationale and experimental analysis could be further strengthened for greater clarity.

**Questions To Address In The Rebuttal:**

Please analyze the model’s performance with additional evaluation metrics and explicitly explain which specific design components enable the proposed modular pipeline to achieve superior performance without requiring multimodal pre-training.

---

> ### Author Response · Authors · 2026-01-24
>
> 1.	C1: The proposed framework shows promise for solid VQA performance at low cost. However, the experimental analysis could be strengthened. Metrics like F1 have limitations in reflecting VQA performance directly—adding indicators such as BLEU-4 would make the assessment more comprehensive.
> We appreciate the suggestion to consider additional metrics. However, BLEU-4 is not well-suited for EndoVis-18-VQA, as the benchmark primarily consists of discrete, closed-set answers (e.g., predefined organ, spatial bins, and interaction states), rather than free-form text generation. In such classification-style VQA settings, BLEU-based metrics are generally inappropriate.
> Instead, similar to relevant studies, we report Accuracy, Recall, and F1, which are standard for this benchmark and directly reflect classification performance across imbalanced answer categories. To further strengthen the manuscript, we have now extended the experimental setup section to include the rationale for choosing EndoVis-18-VQA. Additionally, the discussion section has been extended to discuss class imbalance and per-question-type performance. We also include a discussion in regard to computation cost and latency.
>
>
> 2.	C2: It would also help if the authors clarify which design choices allow the pipeline to work without multimodal training?
>
> We thank the reviewer for the suggestion to clarify this point. Our ability to avoid multimodal training stems from two key design choices. First, we rely on high-quality, task-specific specialist models that are already well established in surgical computer vision (e.g., segmentation and tool–tissue interaction). These specialists provide reliable and structured visual information, removing the need to relearn such representations through end-to-end vision–language training. This is further supported by our oracle experiments (Table 2), where providing ground-truth specialist outputs yields a substantial performance gain.
> Second, we introduce a deterministic abstraction step that converts specialist predictions into structured, human-readable textual descriptions. This allows a frozen LLM to operate purely in the language domain, without any vision–language alignment or domain-specific fine-tuning. Together, these design choices enable strong VQA performance while remaining data-efficient and modular. We have now clarified this more explicitly in the introduction section of the revised manuscript.
>
>
> 3.	C3: Since no LLMs were fine-tuned, it would be valuable to explain the significance of comparing different LLMs in experiments.
> We thank the reviewer for their suggestion. Although we do not fine-tune any LLMs, comparing multiple frozen language models serves both practical and analytical purposes. From a practical perspective, it provides guidance to practitioners with different deployment constraints by illustrating how the proposed pipeline performs across commercial and open-source models of varying sizes. From an analytical perspective, it allows us to assess the robustness of the pipeline to the choice of LLM.
> Notably, our oracle experiments show that once high-quality structured outputs from specialist models are provided, even relatively compact LLMs achieve strong performance, in some cases exceeding prior state-of-the-art results. This indicates that the dominant performance bottleneck lies in perception and abstraction rather than in the capacity of the language model itself. We have clarified this interpretation in the Results section of the revised manuscript.

---

### Official Review · Reviewer_b2XM · 2026-01-13

**Confidence:** 4
**Preliminary Rating:** 4
**Final Rating:** 4

**Summary:**

This paper proposes a pipeline for surgical VQA that explicitly decouples perception from reasoning.
Instead of training an end-to-end VLM, the authors rely on frozen, task-specific vision experts (semantic segmentation and tool–tissue interaction models) to extract structured visual signals.
These signals are then converted via deterministic, hand-crafted heuristics into symbolic textual descriptions, which are finally passed to a frozen LLM to answer questions.
The approach is evaluated on the EndoVis-18-VQA benchmark and shows improvements over prior surgical VQA baselines.

**Strengths:**

It has a couple of strengths:

x. Clear architectural separation with strong motivation. The paper rejects end-to-end multimodal alignment. The decomposition into (vision --> symbolic abstraction --> language reasoning) is well-motivated for surgical domains, where labeled multimodal data is limited but task-specific vision supervision is available.

x. Strong use of specialist vision models. Leveraging segmentation and tool–tissue interaction experts is sensible and empirically justified. The ablation results clearly show complementary strengths: segmentation supports spatial questions, while interaction models support state-related questions.

x. Careful experimental design and ablations. The paper includes segmentation-only vs interaction-only vs combined experts, oracle (ground-truth) upper bounds, and comparison across multiple frozen LLMs. These experiments convincingly demonstrate that performance gains arise from better perception and abstraction, not from LLM fine-tuning.

x. Practical relevance for data-scarce domains.
The method offers a recipe for deploying LLM-based reasoning in domains where multimodal foundation models are infeasible to train or adapt.

**Weaknesses:**

I also identify a few weaknesses:

x. **Limited novelty in the reasoning component**. The LLM is used purely as a black-box text reasoner. No new reasoning mechanism, prompt strategy, or uncertainty handling is introduced. Once the scene description is fixed, most tested LLMs behave similarly under oracle conditions, suggesting limited contribution on the language side.

x. **Heavy reliance on hand-crafted heuristics**.  The abstraction step is entirely rule-based (e.g., spatial partitioning, dilation thresholds, proximity bins). These heuristics are domain-specific, manually designed, and might be sensitive under distribution shift.
Their generalizability beyond EndoVis-style laparoscopic scenes is unclear.

x. **Evaluation task is relatively simple**. EndoVis-18-VQA consists primarily of single-frame, short-horizon questions with discretized answers. The benchmark does not test temporal reasoning oor knowledge-intensive queries.
As a result, the demonstrated “reasoning” may be overstated.

x. **Unclear computational costs of LLM evaluation**. The paper evaluates large models (e.g., Qwen-VL-32B, 72B) but does not clearly state whether inference is local or API-based (i think they are mostly API-based as it's hard to fit them into a single 4090), nor report cost. This weakens claims about practical deployment.

x.**Conceptual proximity to prior symbolic and scene-graph approaches**.
The idea of converting vision outputs into symbolic representations for reasoning is well explored in earlier scene-graph VQA and neuro-symbolic systems. The paper does not sufficiently position itself relative to this literature. More citations and disucssion might be needed.

**Detailed Comments:**

Please see my comments above to address in the rebuttal.

**Justification Of Final Rating:**

The authors addressed most of my concerns in their rebuttal.
Although the methodology novelty is limited from my point of view, it might raise discussion on reasoning for surgical vision at MIDL.
Hence, I vote for weak accept for this work.

**Justification Of The Preliminary Rating:**

My major concern is that the approach is heuristic-heavy and benchmark-dependent. It excels as a practical engineering solution, but does not fundamentally advance our understanding of vision–language reasoning. For details, please see my comments on the weaknesses.

**Questions To Address In The Rebuttal:**

My major concern is that the approach is heuristic-heavy and benchmark-dependent. It excels as a practical engineering solution, but does not fundamentally advance our understanding of vision–language reasoning. For details, please see my comments on the weaknesses.

---

> ### Author Response · Authors · 2026-01-25
>
> 1.	C1: Limited novelty...
> The authors thank the reviewer for their comment. Indeed, the novelty doesn't lie in the internal reasoning mechanisms of the LLM. The comparison across multiple LLMs is included precisely to support this point: once provided with structured outputs from specialist models, even relatively small language models achieve strong performance. This highlights that, in surgical VQA, the primary challenge lies in extracting and abstracting reliable visual information (perception), a task for which specialist models are already well suited, rather than in end-to-end vision-language modeling. By decoupling perception from reasoning and treating the LLM as a black-box consumer of structured facts, our pipeline avoids reliance on scarce multimodal data and expensive multimodal training. The main contribution therefore lies in demonstrating that abstraction-driven modularity with expert-trained specialists is sufficient for competitive performance in data-scarce surgical settings. To make this positioning clearer, we have strengthened the abstract in the revised manuscript to explicitly state the novelty.
>
> 2.	C2: Heavy reliance on ...
> We thank the reviewer for raising this point. The heuristics in our pipeline are intentionally simple, transparent, and task-driven, rather than optimized for a specific dataset. The spatial partitioning and proximity rules operate on generic geometric properties of segmentation masks (e.g., relative location and adjacency) that are common across laparoscopic scenes, rather than encoding EndoVis-18–specific priors. More broadly, the goal of this work is not to propose a fixed set of heuristics, but to demonstrate that abstracting specialist outputs into structured textual representations enables effective downstream reasoning. Importantly, the modular design allows these heuristics to be adapted or extended for new procedures without retraining the language model. To make this clearer, we have emphasized this extensibility in the Discussion and expanded the Limitations section accordingly in the revised manuscript.
>
> 3.	C3:Evaluation task is...
> We opted for EndoVis-18-VQA  as it is a widely used benchmark in surgical VQA and captures two core frame-wise capabilities: spatial understanding (via segmentation) and functional understanding (via instrument–tissue interaction). Importantly, the proposed heuristics are task-driven rather than dataset- or procedure-specific, and can in principle be applied to other surgical scenes with similar visual structure. More broadly, our objective in this work is to first establish the effectiveness of abstraction-driven modularity in a controlled setting before extending to more complex scenarios. The pipeline illustrated in Figure 1 is not inherently limited to single frames and naturally extends to video by incorporating temporal specialists such as phase recognition or action triplet models and aggregating structured outputs over time through the same abstraction mechanism. To make this clearer, we have now clarified this motivation in the Experimental Setup and strengthened the Discussion and Limitations sections in the revised manuscript, leaving temporally richer evaluations for future work.
>
> 4.	C4: Unclear computational ...
> All large language models in our experiments were accessed via external APIs rather than run locally: OpenAI and DeepSeek models were queried through their official APIs, while Qwen-VL models were accessed via OpenRouter. The local workstation was used exclusively for training and inference of the specialist vision models. This distinction is now made explicit in the Experimental Setup section of the revised manuscript. In addition, we have extended Table 2 to report average LLM inference cost and latency per sample, and added a dedicated efficiency paragraph analyzing cost, model scale, and latency. We thank the reviewer for raising this point and refer them to the revised manuscript with highlighted updates.
>
> C5: Conceptual proximity ...
> We thank the reviewer for this comment. While our work shares the high-level idea of using structured intermediates, it differs fundamentally from prior scene-graph and neurosymbolic VQA approaches. Scene-graph methods typically rely on learning or explicitly predicting a graph representation and performing reasoning via graph inference or program execution, often requiring end-to-end multimodal training. In contrast, our approach deliberately avoids learning a symbolic structure: perception is handled by frozen, task-specific surgical specialists, whose outputs are deterministically abstracted into lightweight textual facts, and reasoning is delegated entirely to a frozen LLM. This design shifts the complexity from learned symbolic modeling to modular abstraction, enabling strong performance without multimodal training or graph supervision. To clarify this distinction, we have expanded the related work discussion in the introduction section of the revised manuscript.

---

> > ### Comment · Reviewer_b2XM · 2026-02-01
> > **thank you**
> >
> > The response addressed most of the concerns. Thank you for including the suggestions to updated version.

---

### Author Rebuttal · Authors · 2026-01-25

**Rebuttal:**

We thank the reviewers for their careful reading and constructive feedback, which significantly helped improve the clarity, rigor, and completeness of the manuscript. We have addressed all comments and questions in detail in the individual responses.

In the revised manuscript, we include several highlighted (blue text font) updates and additions. These include minor clarifications and refinements throughout the paper, as well as new experimental analyses addressing computational efficiency, sensitivity to heuristic parameters, and the relationship between specialist model performance and VQA performance. We also extend the Results and Discussion sections with dedicated analyses of inference cost, latency, and practical deployment considerations, including clinical feasibility.

In addition, we strengthen the discussion and limitations to clarify the scope, extensibility, and generalizability of the proposed framework, and expand the related work to better position the approach relative to prior symbolic and scene-graph methods. Where requested, we also include qualitative examples and supplementary analyses to further illustrate design choices.

We believe these revisions address the reviewers’ concerns and substantially strengthen the manuscript, and we thank the reviewers again for their time and valuable input. Please find attached the revised manuscript and an additional figure corresponding to comment 3 from Reviewer K57y, illustrating the limitations of overlaying specialist outputs for VLM prompting.

**Supporting Material:**

/attachment/30fec000bc6f5fb0b4bb017d2f05f16a133c9593.zip

---

### Meta-Review · Area_Chair_Uzep · 2026-02-09

**Recommendation:** Accept (Poster)
**Confidence:** 5

**Metareview:**

Three reviewers provided **weak accept**, and one reviewer improved their rating from weak reject to **borderline**.

The final decision reflects the reviewers’ overall evaluations. The authors are encouraged to carefully address the remaining concerns and incorporate key clarifications from the rebuttal into the final manuscript.

The primary outstanding concern is that the proposed method depends on manual context engineering, which may limit its generalizability to other types of surgical videos.

---

### Decision · Program_Chairs · 2026-02-13

Accept (Poster)